# Plasma Concentrations of Short-Chain Fatty Acids in Active and Recovered Anorexia Nervosa

**DOI:** 10.3390/nu14245247

**Published:** 2022-12-09

**Authors:** Jingjing Xu, Rikard Landberg, Catharina Lavebratt, Cynthia M. Bulik, Mikael Landén, Ida A. K. Nilsson

**Affiliations:** 1Department of Molecular Medicine and Surgery, Karolinska Institutet, 171 76 Stockholm, Sweden; 2Center for Molecular Medicine, Karolinska University Hospital, 171 76 Stockholm, Sweden; 3Department of Biology and Biological Engineering, Food and Nutrition Science, Chalmers University of Technology, 412 96 Gothenburg, Sweden; 4Centre for Eating Disorders Innovation, Karolinska Institutet, 171 77 Stockholm, Sweden; 5Department of Medical Epidemiology and Biostatistics, Karolinska Institutet, 171 77 Stockholm, Sweden; 6Department of Psychiatry, University of North Carolina at Chapel Hill, Chapel Hill, NC 27599, USA; 7Department of Nutrition, University of North Carolina at Chapel Hill, Chapel Hill, NC 27516, USA; 8Department of Psychiatry and Neurochemistry, Institute of Neuroscience and Physiology, The Sahlgrenska Academy at the University of Gothenburg, 405 30 Gothenburg, Sweden

**Keywords:** anorexia nervosa, short-chain fatty acids, plasma, BMI

## Abstract

Anorexia nervosa (AN) is one of the most lethal psychiatric disorders. To date, we lack adequate knowledge about the (neuro)biological mechanisms of this disorder to inform evidence-based pharmacological treatment. Gut dysbiosis is a trending topic in mental health, including AN. Communication between the gut microbiota and the brain is partly mediated by metabolites produced by the gut microbiota such as short-chain fatty acids (SCFA). Previous research has suggested a role of SCFA in weight regulation (e.g., correlations between specific SCFA-producing bacteria and BMI have been demonstrated). Moreover, fecal SCFA concentrations are reported to be altered in active AN. However, data concerning SCFA concentrations in individuals who have recovered from AN are limited. In the present study, we analyzed and compared the plasma concentrations of seven SCFA (acetic-, butyric-, formic-, isobutyric-, isovaleric-, propionic-, and succinic acid) in females with active AN (*n* = 109), recovered from AN (AN-REC, *n* = 108), and healthy-weight age-matched controls (CTRL, *n* = 110), and explored correlations between SCFA concentrations and BMI. Significantly lower plasma concentrations of butyric, isobutyric-, and isovaleric acid were detected in AN as well as AN-REC compared with CTRL. We also show significant correlations between plasma concentrations of SCFA and BMI. These results encourage studies evaluating whether interventions directed toward altering gut microbiota and SCFA could support weight restoration in AN.

## 1. Introduction

Up to 4% of females and 0.3% of males are estimated to be diagnosed with the psychiatric disorder anorexia nervosa (AN) during their lifetime [1], a disorder with both high mortality (up to 10%) and relapse rates [2,3,4], illustrative of its severity. Core features of AN are persistent restriction of food intake resulting in severe underweight, fear of gaining weight, persistent behaviors that interfere with weight gain, and a distorted body image or lack of recognition of the seriousness of the low weight [5,6]. Heritability of AN has been established by several twin studies, where 58–70% of the variance in liability has been found to be explained by additive genetic factors [7]. The latest genome-wide association study (GWAS) identified eight loci associated with AN, as well as significant genetic correlations with several other psychiatric disorders and metabolic traits [8]. Environmental and neurobiological factors also contribute to the development and/or maintenance of AN [5]. However, we still lack a complete understanding of the (neuro)biology of AN, and importantly, there are no medications indicated for or effective in its treatment. Defining molecular mechanisms involved in extreme weight and behavioral dysregulation has the potential to identify new pharmacological targets and new interventions to aid in weight restoration and recovery.

Several studies have documented a dysbiosis of gut microbiota in individuals with AN who experience drastically altered diets and prolonged low energy intake [9,10,11,12,13,14]. In addition, fecal transplantation of the gut microbiome derived from AN patients has been suggested to impede weight gain in female mice [15], and fecal microbiota transplantation to an anorexia patient with previous unsuccessful attempts to maintain a healthy body weight resulted in weight gain [16]. Moreover, obese mice possess a microbiota with an increased capacity to extract energy from the diet [17], and microbes in their gut appear to mediate the metabolic benefits associated with calorie restriction [18]. Furthermore, the existence of specific bacteria decreases the rate of diet-induced weight loss in pet dogs [19]. In humans, a longitudinal study found baseline gut microbiota to be the most important predictor of individual diet-induced weight-loss trajectories [20].

The gut communicates with other organs including the brain (i.e., the gut-brain axis) via metabolites produced by the microbiota. Upon microbial fermentation of carbohydrates, such as dietary fibers and resistant starch, short-chain fatty acids (SCFAs) are produced. Lower fecal concentrations of the SCFAs butyrate and propionate, and occasionally acetate, have previously been reported in individuals with active AN [9,10,21,22]. In addition, a systematic review by di Lodovico et al. [23] concluded that a decrease of butyrate-producing species such as *Roseburia*, as well as an increase of mucine-degrading species such as *Akkermansia*, may represent hallmarks of the gut microbiota alterations in AN. Interestingly, *Akkermansia* supplementation has based on its beneficial effects on health and metabolism been suggested as an intervention for obesity [24]. Moreover, Lodovico et al. reported a positive correlation between butyrate-producing bacteria and BMI [23]. Lastly, the microbiota diversity, both beta and alpha diversity, has been reported to be reduced in AN compared to healthy controls [11,12,13,14,22]. Limited data indicate that perturbations in the gut microbiome and SCFA levels in AN do not normalize with weight gain [14]. However, to date, no consensus exists as to what extent the gut dysbiosis in AN is confined to the active illness state or if it persists after recovery. Clarifying the role of SCFAs coupled to the gut microbiome in weight regulation and AN is therefore of considerable scientific and clinical interest. Accordingly, we estimated plasma concentrations of seven SCFAs (i.e., acetic-, butyric-, formic-, isobutyric-, isovaleric-, propionic-, and succinic acid) in females with active AN or recovered from AN (AN-REC), as well as healthy female controls.

## 2. Materials and Methods

### 2.1. Population and Study Design

This cross-sectional study included 327 females all identified from the Swedish sample of the Anorexia Nervosa Genetics Initiative (ANGI-SE). Details on the recruitment procedure have been described by Thornton et al. [25]. The inclusion criteria for the AN group were female patients, at least 18 years of age, meeting DSM-IV criteria for AN [26] except for amenorrhea, and a minimum of one year since AN onset *(n* = 109). For AN-REC, the inclusion criteria were a history of DSM-IV AN diagnosis followed by weight restoration (BMI > 20 kg/m^2^), plus no eating disorder behaviors for at least a year (*n* = 108). One year was chosen based on the recovery literature since it is the most common minimum symptom-free duration required to be considered recovered, and ensures to some extent, that relapse is less likely [27]. The age-matched normal-weight female controls reported no history of disordered-eating behavior (CTRL, *n* = 110). Stratified analyses of the two AN groups were carried out exploring subtypes; AN with binge (AN-B, *n* = 44, AN-REC-B *n* = 70) as defined by episode(s) of binge eating with loss of control, while complete absence of such episodes was defined as AN without binge eating (AN-noB, *n* = 52, AN-REC-noB, *n* = 30). We also performed a stratified analysis of active AN, as well as AN-REC, with documented episodes of laxative use (AN-LAX, *n* = 22, AN-REC-LAX, *n* = 23) compared with without laxative use (AN-noLAX, *n* = 85, AN-REC-noLAX, *n* = 83), as well as with documented self-induced vomiting (AN-VOM, *n* = 42, AN-REC-VOM, *n* = 55) compared to without vomiting (AN-noVOM, *n* = 65, AN-REC-noVOM, *n* = 51). See Table 1 for detailed characteristics of the study participants. The study followed the principles of the Declaration of Helsinki and was approved by the Regional Ethics Review Board in Stockholm. All participants gave written informed consent.

### 2.2. Blood Sampling

Blood was collected using EDTA tubes at a hospital or a lab near the participant, sent to Karolinska Institutet Biobank with overnight mail, and processed upon arrival. After centrifugation, plasma samples were stored at −80 °C pending analysis at the Chalmers University of Technology.

### 2.3. LC-MS

Nine SCFAs (acetic-, butyric-, caproic-, formic-, isobutyric-, isovaleric-, propionic-, succinic-, and valeric acid) were analyzed in EDTA plasma by liquid chromatography-mass spectrometry (LC-MS) according to a method described previously [28] with some modifications. All reference compounds, except for 13C6-3NPH (custom synthesized by IsoSciences Inc. King of Prussia, PA, USA), the internal standard for all SCFAs, solvents, and reagents were purchased from Sigma-Aldrich. To avoid SCFA contamination, hyper grade LC-MS water and MeOH (Lichrosol) were used, and all reagents and solvents were used for a maximum of 5 days and then replaced. In brief, plasma (10 μL) was incubated with 75% methanol (60 μL) and mixed with 200 mM 3-NPH (60 μL) and 120 mM EDC-6% pyridine (10 μL) at ambient temperature for 45 min under gentle shaking. The reaction was quenched by the addition of 200 mM quinic acid (10 μL) at gentle shaking at ambient temperature for 15 min. The samples were centrifuged at 15,000× *g* for 5 min and the supernatant was moved to a new tube. The sample was made up to 1 mL by 10% methanol in water and again centrifuged at 15,000× *g* for 5 min. In total 100 μL of the derivatized (12C) sample was mixed with 100 μL of labeled (13C) internal standard. Samples were analyzed by a 6500+ QTRAP triple-quadrupole mass spectrometer (AB Sciex, 11432 Stockholm, Sweden) which was equipped with an APCI source and operated in the negative-ion mode. A Phenomenex Kinetix Core-Shell C18 (2.1, 100 mm, 1.7 um 100Å) UPLC column with SecurityGuard ULTRA Cartridges (C18 2.1 mm ID) (changed at regular intervals) was used for separation of the analytes. The column was backflushed for 60 min between each batch to ensure good chromatographic separation. LC-MS grade water (100% solvent A) and acetonitrile (100% solvent B) were the mobile phases for gradient elution. The column flow rate was 0.4 mL/min and the column temperature was 40 °C, the autosampler was kept at 5 °C. The gradient started at 0.5% B (held for 3 min), 2.5% B ramping linearly to 17% B at 6 min, then to 45% B at 10 min, and 55% B at 13 min. Followed by a flush (100% B) and recondition (0.5% B), a total runtime of 15 min. The multiple reaction monitoring (MRM) transitions were optimized for the analytes one by one by direct infusion of the derivatives containing 50 mM of each fatty acid. The Q1/Q3 pairs were used in the MRM scan mode to optimize the collision energies for each analyte, and the two most sensitive pairs per analyte were used for the subsequent analyses. The retention time window for the scheduled MRM was 1 min for each analyte. For the two MRM transitions per analyte, the Q1/Q3 pair that showed the higher sensitivity was selected as the MRM transition for quantitation. The other transition acted as a qualifier for verification of the identity of the compound. Linear, eight-point calibration curves were prepared for each reference compound and used for quantification. The intra-and inter-batch variations were calculated based on the inclusion of three different QC samples with different concentrations in triplicates in each batch across seven batches. The mean intra-batch variation was between 3–11% for all SCFA except valeric acid (>15%), for which the inter-batch variation also was high (>40%), and subsequently was not included in further analyses. Caproic acid was also excluded since unpublished data from our lab on another cohort of plasma show a low run-rerun correlation for this low abundant SCFA. The inter-batch variation was controlled for by normalizing the sample values with the QC values. The normalization factor for each analyte per batch was calculated by the mean of QC values of the individual batch/mean of the total QC values from all seven batches. The statistical analyses were performed on normalized data.

### 2.4. Statistical Analyses

Demographic and clinical characteristics of AN, AN-REC, and CTRL groups were analyzed using descriptive statistics. Group differences in plasma concentrations of the seven SCFAs (acetic-, butyric-, formic-, isobutyric-, isovaleric-, propionic-, and succinic acid) in AN, AN-REC, and CTRL were analyzed using the Kruskal-Wallis test since the plasma concentrations of analytes were not normally distributed, followed by posthoc Dunn’s test with Bonferroni correction to evaluate pairwise comparisons. Group differences in plasma concentrations of SCFA in AN-B vs. AN-noB, AN-REC-B vs. AN-REC-noB, AN-LAX vs. AN-noLAX, AN-REC-LAX vs. AN-REC-noLAX, AN-VOM vs. AN-noVOM, as well as AN-REC-VOM vs AN-REC-noVOM were tested using the nonparametric Mann–Whitney U test.

Associations between SCFA concentrations and age, BMI, as well as years since AN onset, were assessed using linear regression models. If a significant correlation between SCFA levels and age was found, we would adjust for age in further analyses for this SCFA. Spearman correlation revealed that the plasma concentrations of five out of seven SCFA correlate with each other (Appendix A). We thus adjusted *p*-values to account for the three independent tests, and an alpha level of 0.017 was therefore considered statistically significant. All statistical analyses were performed using R programming language version 4.1.0 (including package emmeans). Graphs were made using the ggplot2 package from R.

## 3. Results

The demographic and clinical characteristics of the study population are summarized in Table 1. We observed significant differences in; BMI AN vs. AN-REC (*p* = 1.69 × 10^−30^), and AN vs. CTRL (*p* = 1.67 × 10^−44^), minimum BMI AN vs. AN-REC (*p* = 8.02 × 10^−19^), as well as length of amenorrhea AN vs. AN-REC (*p* = 3.29 × 10^−6^). Psychiatric comorbidities of the study participants are summarized in Appendix A, stratified according to ICD-10, International Statistical Classification of Diseases and Related Health Problems, 10th Revision.

We observed no effect of age at sampling on normalized plasma concentrations of any of the seven SCFAs analyzed, thus further analyses were not adjusted for age. Analyses of group differences showed that plasma concentrations of butyric-, isobutyric-and isovaleric acid were significantly lower not only in AN compared with CTRL, but also in AN-REC compared with CTRL (Figure 1).

Plasma isobutyric acid concentration correlated positively with overall BMI (Figure 2B). When each of the three groups was investigated separately, we observed a significant negative correlation between BMI and plasma SCFA concentrations for succinic-and propionic acid in the AN group (Figure 2D,G). No correlations between BMI and SCFA concentrations were observed in AN-REC or CTRL group (Figure 2). The slopes of the correlations between BMI and SCFA concentrations (Figure 2, Appendix A) differed significantly between AN and CTRL (*p* = 0.004), as well as between AN and AN-REC (*p* = 0.005), for propionic acid. The same was seen for succinic acid, AN vs CTRL (*p* = 0.005) and AN vs AN-REC (*p* = 0.007). No significant differences were found between the AN-REC and CTRL groups with respect to correlations between BMI and SCFA concentrations (Appendix A). We detected no significant correlation between years since AN onset and SCFA plasma concentration in the AN group.

We detected a significantly higher plasma concentration of succinic acid in AN-LAX compared with AN-noLAX. No other significant differences were seen for neither AN nor AN-REC with regards to laxative use (Figure 3).

Significantly reduced butyric acid was seen for AN-VOM compared with AN-noVOM. No other significant differences were seen for neither AN nor AN-REC with regards to self-induced vomiting (Figure 4).

We also see a significant difference with reported binge eating in the AN-REC group; lower plasma isovaleric acid in AN-REC-B compared with AN-REC-noB (Figure 5), whereas no significant differences were seen in active AN patients with or without reported binge eating (Figure 5).

## 4. Discussion

We sampled plasma from 109 females with active AN, 108 females who had recovered from AN, and 110 healthy controls and evaluated concentrations of the microbiota metabolites SCFAs. We successfully analyzed seven SCFAs using a novel high-throughput LC-MS method and found a significantly lower concentration of butyric acid in plasma from individuals actively ill as well as recovered from AN, compared with CTRL. Although this partially aligns with some prior results—lower fecal levels of butyrate and butyrate-producing species have previously been reported in active AN [9,10,22,23]—it has to our knowledge not been reported previously in AN plasma nor been documented to persist after recovery from AN, in neither plasma nor feces. In addition, we observed reduced plasma concentrations of isobutyric-and isovaleric acid in both AN groups compared to CTRL. Caloric restriction of rats has been shown to reduce the microbial enzymes responsible for butyrate as well as acetate synthesis, while increasing the ones for propionate synthesis, thus only partly aligning with the results presented here [29]. Taken together, the lower concentration of butyric, isobutyric, and isovaleric acid in both AN and AN-REC compared with CTRL indicates that the changed SCFA profile is not a direct effect of weight loss as has been discussed previously [30] since AN-REC per definition were weight recovered for at least one year prior to blood sampling. The median minimal BMI during AN is lower for the AN group compared with the AN-REC group, and the median length of amenorrhea is longer in AN vs AN-REC, which could indicate more severe and enduring cases in the former group. Nonetheless, we see the reductions in SCFA also in the latter group when compared with CTRL.

Contrary to our results, isobutyrate has previously been reported to be increased in fecal samples from AN patients both at hospital admission and the end of the hospital stay when weight was partially recovered [14]. We also report a significant positive correlation between BMI overall and isobutyric concentrations. When evaluating the correlations between BMI and SCFA concentrations stratified by the three groups, we observed significant negative correlations between BMI and plasma SCFA concentrations for propionic-and succinic acid in the AN group only. This indicates a specific association between these gut microbiota metabolites in the extremely low BMI/AN condition. The lower fecal propionate and acetate previously reported in AN [9,10] were not reflected in lower plasma concentrations in our study. On the other hand, the propionate-producing microbe *Akkermansia muciniphila*, a known modulator of metabolism, has been reported to be increased in fecal samples from individuals with AN, whereas reduced levels have been seen in individuals with obesity [23,31]. It is, however, important to mention that there does not seem to be any clear correlation between fecal and plasma SCFA concentrations, likely reflective of SCFA absorption efficiency and in vivo utilization of the acids [30,32,33], thus our results in plasma cannot be directly compared with previous results on fecal samples. In fact, it has been suggested that upon weight loss, SCFAs are utilized as an energy source in peripheral tissues as well as in the colon itself [17], which thus would reduce the concentrations in both feces and plasma. Observational studies are consistent with this theory reporting lower SCFA concentrations in feces from lean individuals [34,35]. Nevertheless, we report here reduced concentrations in AN-REC, which supposedly should not be in a state of negative energy balance and weight loss. On the other hand, a leaky gut theory has been discussed in AN [23,36], which rather would favor increased SCFA concentrations in plasma. Increased mucin-degradation [14] and decreased trophic effect on enterocytes [9] documented in AN, combined with the knowledge that stress, high cortisol, and excessive physical activity, all commonly seen in AN, could contribute to such an increased permeability [36]. In addition, reduced expression of tight junctions and a thinner intestinal wall were reported in an animal model of anorexia [37]. Studies evaluating gut permeability in AN are however conflicting [13,38].

We also report a significantly higher concentration of succinic acid in AN with reported use of laxatives compared with those not reporting such use. This difference is seen despite the fact that we were not able to control the time since the last laxative use episode, the frequency of usage, or the type of laxative. However, laxatives have been shown in mice to cause long-term changes in gut microbiota (i.e., a new steady state in the microbiota, and extinction of key taxa) [39]. Moreover, we report significantly reduced butyric acid in AN with reported self-induced vomiting compared with those not reporting it. No differences were seen for AN-REC with or without history of laxative use or self-induced vomiting. Furthermore, we report significantly lower plasma isovaleric acid concentration in AN-REC with reported binge eating compared with non-binge eating AN-REC. However, no differences were identified when evaluating reported binge eating in active AN patients.

The potential causality between deviant SCFA plasma concentrations and AN cannot be elucidated with the present design. However, preclinical and clinical studies, including fecal transplantation, support the role of the gut microbiome and SCFA in weight regulation [15,16,19,20]. Additional evidence for the role of SCFA in energy homeostasis [40,41] includes their ability to stimulate food intake by inhibiting gut hormones, glucagon-like peptide-1 (GLP-1), and peptide YY (PYY) [42,43]. Fermentable carbohydrates, as well as intraperitoneal administration of acetate, have also been reported to increase signaling in the food intake regulating centers of the hypothalamus [44,45,46]. Targeted delivery of propionate to the colon has been reported to prevent weight gain in overweight adult humans [47]. Animal studies have demonstrated that dietary supplementation with specific SCFA protects against diet-induced obesity via direct effects on the adipose tissue and gut hormones [43,48]. Therefore, we speculate that the aberrant SCFA profile in AN may interfere with the systems regulating food intake and weight, thereby supporting underweight and negative energy balance.

Other routes by which SCFA could influence weight and food intake regulation are via hypothalamic inflammation and microglia. SCFAs—in particular butyrate—here shown to be lower in AN and AN-REC plasma, are crucial for microglia maturation and activation. These brain cells appear to require continuous stimulation from the gut microbiota to remain mature [49,50]. Microglia are supportive glial cells in the CNS that, among others, respond to brain injury and are involved in immune activation [51]. We previously reported activation of microglia selectively in the food intake regulating systems in the hypothalamus of an animal model of AN, the *anx/anx* mouse [52]. Similarly, hypothalamic inflammation, including activation of microglia and astroglia, has been documented in diet-induced obesity of rodents as well as in humans with obesity prior to the onset of weight gain [53,54]. Recent data show that the gut microbiota regulates western diet-induced hypothalamic inflammation and microglia maturation via a GLP-1R-dependent mechanism in astroglia [55]. In addition, microglial activation has been documented in several additional psychiatric disorders [56,57,58], whereas deviations in SCFA have been documented in autism spectrum disorder [59,60]. Thus, taken together these findings support the role of SCFA-induced hypothalamic microglia activation involved in weight and food intake regulatory aspects of AN.

The main limitation of our study is the cross-sectional design, which precludes inferences about causality. Another limitation is the blood sampling procedure, as blood samples were sent via overnight mail before being processed. Even if this may have influenced the absolute SCFA concentration [61], it is unlikely to have affected group differences since all samples were collected and handled in this same way. We did in fact detect plasma concentrations of butyrate higher than previously observed in any dataset analyzed in the laboratory (data not shown), while formate was somewhat lower than typically observed [62]. However, this was true also for the healthy controls. Plasma butyrate as well as formate concentrations, in similar ranges as reported here, have previously been reported [63,64,65,66]. In addition, the sampling was not standardized for a specific time of the day, and participants were not instructed to be in a fasted or fed state. Furthermore, valeric and caproic acid were both excluded since the quality control of LC-MS analysis indicated high inter-batch variation and low run-rerun correlation in another cohort originating from our laboratory (Lavebratt et. al., unpublished), respectively, likely due to the very low concentrations of these SCFA in plasma. However, isovaleric acid was also low in plasma, and these results thus should be interpreted with caution. Another limitation is that fecal concentrations of SCFA and microbiota taxonomy were not measured in combination with the plasma concentrations. In addition, we required a one-year duration of recovery from AN, which is consistent with many definitions of recovery but it may not be adequate for the normalization of the microbiota [67]. It is also possible that the dietary habits of the AN-REC group still differ from individuals without any history of an eating disorder [68] in ways that could influence the gut microbiota (e.g., the amount of ingested dietary fibers and resistant starch). Unfortunately, we do not have access to dietary records. Both alcohol intake and smoking are also known to affect the gut microbiota and SCFA production [69,70,71], but such data are not available. Neither do we have data on the individual’s pharmacological treatment at time of sampling. In addition, we do not have data on the duration of the disorder for the AN-REC group. However, since the medians for age at sampling, age at onset, and years since AN onset are the same in the two groups, combined with the inclusion criteria of a minimum one year since recovery for AN-REC, we can conclude that the illness duration is at least a year shorter in the AN-REC group. We do not have information on the exact time since recovery and did not set a maximum time for this parameter.

Future research should evaluate plasma concentrations of SCFA in AN-REC with a longer defined time since recovery. The optimal design would be a longitudinal study that followed individuals over the course of treatment through recovery that measured SCFA concentrations in plasma and feces combined with dietary records to decipher the effect of nutritional recovery post AN on the gut microbiota and its metabolites. Lastly, it would also be valuable to further explore the effects of SCFA on glial cells and neurons of interest for the AN pathology (i.e., neurons involved in energy homeostasis).

## 5. Conclusions

Here, we report significantly lower plasma concentrations of butyric-, isobutyric-, and isovaleric acid in individuals with active AN—as well as in individuals who had recovered from AN for at least one year—compared with healthy controls. This combined with the knowledge that SCFA can intervene with several food intake and body weight regulating mechanisms, suggests an opportunity to test the augmentation of standard treatment with these specific metabolites as a potential route to support, enhance, and maintain weight recovery in AN [72]. Although all novel and defensible interventions for AN are deserving of study, such precision approaches may provide an alternative to more invasive fecal transplantations, which have been tested with some success, but are not acceptable to all patients and carry some risk [16,73]. Given the documented poor outcome and absence of any effective medications for the treatment of AN, novel approaches to enhance treatment with strategies that are acceptable to patients are of considerable urgency.

## Figures and Tables

**Figure 1 nutrients-14-05247-f001:**
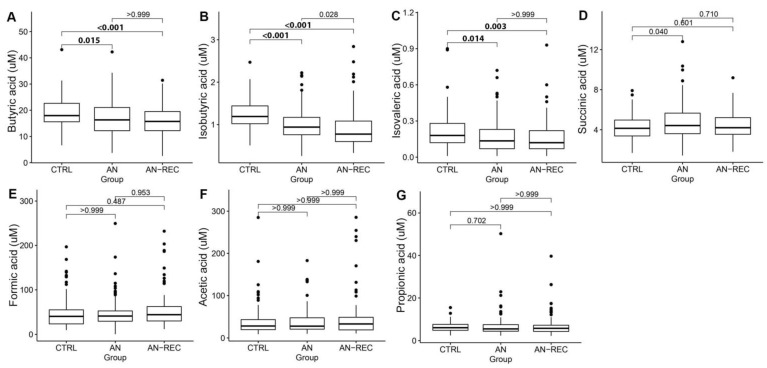
Box plots illustrate group differences in adjusted plasma concentrations of SCFA; butyric-(**A**), isobutyric-(**B**), isovaleric-(**C**), succinic-(**D**), formic-(**E**), acetic-(**F**) and propionic acid (**G**), in active anorexia nervosa (AN), recovered AN (AN-REC), and normal-weight controls (CTRL). The median is shown as a straight line and the box denotes the interquartile range. *p* < 0.017 is considered significant.

**Figure 2 nutrients-14-05247-f002:**
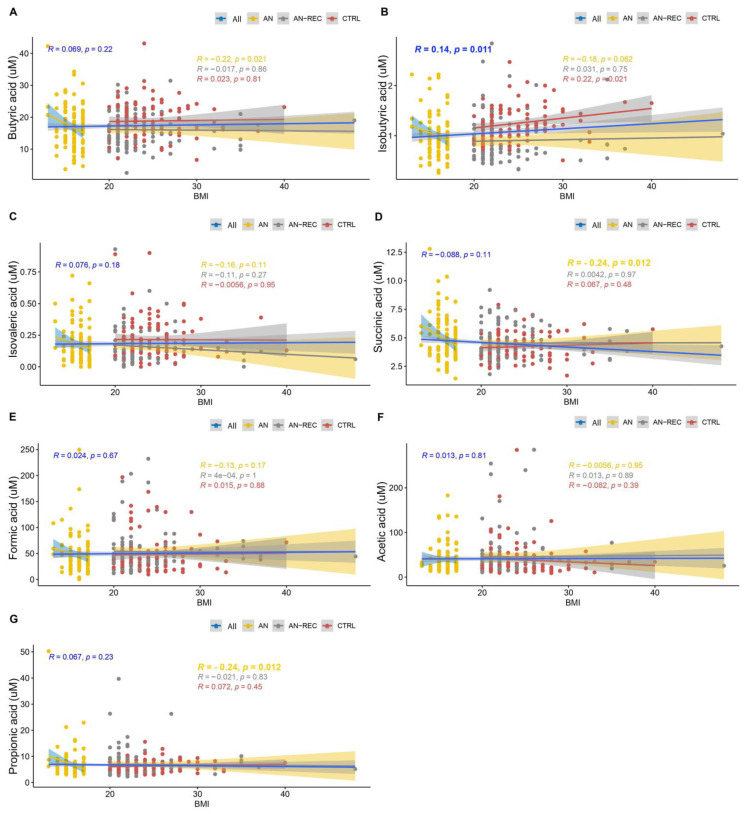
Graphs show correlations between SCFA plasma concentrations; butyric-(**A**), isobutyric-(**B**), isovaleric-(**C**), succinic-(**D**), formic-(**E**), acetic-(**F**) and propionic acid (**G**), and BMI. Lines correspond to medians. Blue lines correspond to the correlation over the whole BMI range, yellow ones to the correlation for the AN group, grey AN-REC, and red CTRL. The shaded line around each linear fit line represents a 95% confidence interval. *p* < 0.017 is considered significant.

**Figure 3 nutrients-14-05247-f003:**
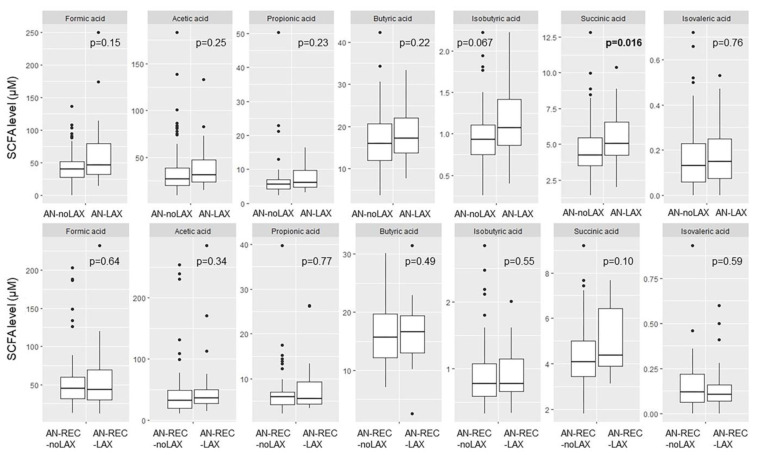
Box plots illustrate group differences in adjusted plasma concentration of SCFA in active and recovered AN with (AN-LAX, AN-REC-LAX) and without (AN-noLAX, AN-REC-noLAX) documented episodes of laxative use. The median is shown as a straight line and the box denotes the interquartile range. *p* < 0.017 is considered significant.

**Figure 4 nutrients-14-05247-f004:**
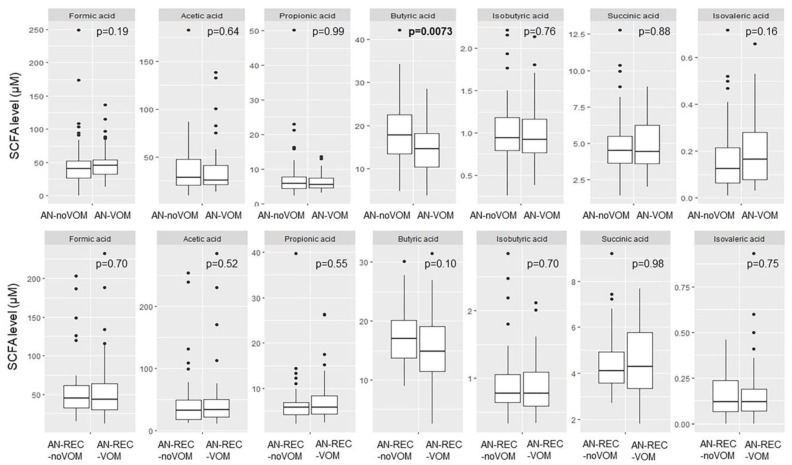
Box plots illustrate group differences in adjusted plasma concentration of SCFA in active and recovered AN with (AN-VOM, AN-REC-VOM) and without (AN-noVOM, AN-REC-noVOM) documented self-induced vomiting. The median is shown as a straight line and the box denotes the interquartile range. *p* < 0.017 is considered significant.

**Figure 5 nutrients-14-05247-f005:**
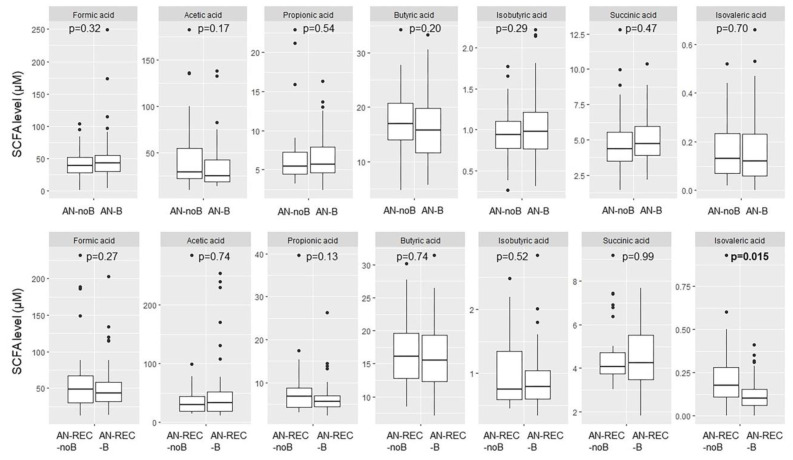
Box plots illustrate group differences in adjusted plasma concentration of SCFA in active and recovered AN (AN-REC) with (AN-B, AN-REC-B) vs without (AN-noB, AN-REC-noB) reported binge eating. The median is shown as a straight line and the box denotes the interquartile range. *p* < 0.017 is considered significant.

**Table 1 nutrients-14-05247-t001:** Sex, age, BMI, years since AN onset, length of amenorrhea, binge behavior, laxative usage, and self-induced vomiting of the study participants by group.

Characteristics	AN	AN-REC	CTRL
n	109	108	110
Female (%)	100	100	100
Age at sample (median years [IQR])	26 [24.0–31.0]	26 [24.0–30.25]	26 [24.0–31.0]
Age of first AN onset (median years [IQR])	16 [14.0–19.0]	16 [14.0–19.0]	
BMI at sampling (median kg/m^2^ [IQR])	16 [15.2–16.6]	22 [20.8–24.2]	23 [22.9–26.0]
Minimum BMI during AN (median kg/m^2^ [IQR])	13.7 [12.5–14.5]	16.5 [15.0–17.8]	
Years since AN onset (median [IQR])	10 [6.0–15.0]	10 [6.0–14-.0]	
Length of amenorrhea (median years [IQR])	4 [2.0–8.0]	1.5 [0.5–3.0]	
Subtype (n);			
With binge-eating	44	70	
Without binge-eating	52	30	
Never laxative use	85	83	
Ever laxative use	22	23	
Never self-induced vomiting	65	51	
Ever self-induced vomiting	42	55	

AN, anorexia nervosa; AN-REC, recovered from anorexia nervosa; CTRL, healthy controls; IQR, interquartile range.

## Data Availability

The data presented in this study are available on request from the corresponding author. The data are not publicly available due to sensitive personal data needed to be stored in accordance with General Data Protection Regulation (GDPR).

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
