# Peer review of "Plasma Concentrations of Short-Chain Fatty Acids in Active and Recovered Anorexia Nervosa"

_nutrients, 2022, doi:10.3390/nu14245247_

Round 1

Reviewer 1 Report

The topic of the article is very important, considering the increasing frequency and mortality of AN.

I suggest some changes and answer some questions. Something is wrong with the authors' titles. There is an unnecessary ‘and’ in the list of authors. Is it necessary to indicate the authors' titles? The Materials and methods correspond to the objective, their description requires some minor clarifications. -          Were the members of CTRL group identified from ANGI-SE? -          Why does the sum of the number of participants in the subgroups not match the number of participants in the entire group? ( i.AN-B, n=44, AN-noB, n=52, AN n=109; AN-REC-B n=70, AN-REC-noB, n=30, AN-REC n=108; etc. ) -          Why there are no subtypes AN-REC-LAX, AN-REC-noLAX, AN-REC-VOM and AN-REC-noVOM? -          Line 90-91: „The inclusion criteria for the AN group were female patients, at least 18 years of age, meeting DSM-IV criteria for AN except for amenorrhea”.
In contrast, amenorrhoea is also mentioned in Table 1 and in line 240.
In the Results -          The header of Table 1 does not mention amenorrhea and vomiting. -          In my opinion, it is unnecessary to provide the SCFA concentrations in Table 2 and Fig 1 A,C,E,G,I,K also illustrated.  Table 2 would be enough, and then the correlation graphs could be larger. So now they are unfortunately very small, hard to read. -          Significance marks are missing from Table 2. -          Is not 0.040 in Fig1G significant? -          It would be worthwhile to represent all the significant results in Tables or Figs in the main text and not in the Supplementary Materials. (line 204-209) -          Unfortunately, I could not open the link of Supplementary Materials. -          Does the duration of the AN or the time since recovery affect the results? -          Were there any significant differences in the slopes of the correlations between BMI and SCFA concentrations between the AN subtypes? The Discussion is very detailed and thorough. Nice work.  The list of References is extensive, not all journal names are abbreviated.

Author Response

The topic of the article is very important, considering the increasing frequency and mortality of AN.

I suggest some changes and answer some questions. 

Response: We wish to thank the reviewer for taking the time to review and improve our manuscript.

Something is wrong with the authors' titles. There is an unnecessary ‘and’ in the list of authors.

Response: Thanks for noticing. We removed it.

Is it necessary to indicate the authors' titles? 

Response: We removed the titles.

The Materials and methods correspond to the objective, their description requires some minor clarifications.          

- Were the members of CTRL group identified from ANGI-SE?        

Response: Yes, they were. We clarified this in the materials and methods (line 93).

  - Why does the sum of the number of participants in the subgroups not match the number of participants in the entire group? ( i.AN-B, n=44, AN-noB, n=52, AN n=109; AN-REC-B n=70, AN-REC-noB, n=30, AN-REC n=108; etc. ) 

Response: This discrepancy in numbers stems from that some participants have chosen not to respond to the questions in the ANGI survey used to define the subgroups.

- Why there are no subtypes AN-REC-LAX, AN-REC-noLAX, AN-REC-VOM and AN-REC-noVOM?         

Response: Since we aimed to explore the more immediate or direct effect of the two exposures on SCFA concentration in plasma, we choose to only explore it in active AN. The information we have on laxative use and self-induced vomiting originates from the period during the active disorder. In particular laxative use, we expected to have an effect on the gut microbiota, and thus SCFA concentrations, within the time close after the exposure. However, based on the reviewers comment we realise the value of exploring these two exposures also as proxies of the biology of the restrictive subtype of AN, and have thus included analyses of laxative use and self-induced vomiting also in the AN-REC group. No significant differences were seen in SCFA concentrations between AN-REC with or without laxative/vomiting (Figures 3 & 4, line 253-255; 272-274; 342-343).

Line 90-91: „The inclusion criteria for the AN group were female patients, at least 18 years of age, meeting DSM-IV criteria for AN except for amenorrhea”. In contrast, amenorrhoea is also mentioned in Table 1 and in line 240. 

Response: This definition of AN was set in order to match the DSM-V criteria for AN, which does not include amenorrhea. Thus, amenorrhea is no longer a criterion for AN diagnosis in the latest version of DSM. We did however find it interesting to explore if the length of amenorrhea, in those individuals for which this occurs, influences the plasma concentrations of SCFA. Individuals without amenorrhea were counted as 0 years.

In the Results 

-  The header of Table 1 does not mention amenorrhea and vomiting.      

Response: Thanks for noticing. We have added this.    

- In my opinion, it is unnecessary to provide the SCFA concentrations in Table 2 and Fig 1 A,C,E,G,I,K also illustrated.  Table 2 would be enough, and then the correlation graphs could be larger. So now they are unfortunately very small, hard to read

Response: We thank the reviewer for the suggestion. We have now excluded table 2, and split figure 1 into two figures; Fig.1. box plots of SCFA concentrations, and Fig. 2 correlation graphs, in order to make them easier to read.  

-  Significance marks are missing from Table 2. 

Response: Thanks for noticing. We have, as written above, removed table 2 and present these results as Figure 1 only. Significance is in this figure clearly marked as p-values.

-   Is not 0.040 in Fig1G significant? 

Response: No since we adjust for multiple testing. See line 180-181: “We thus adjusted p-values to account for the three independent tests, and an alpha level of 0.017 was therefore considered statistically significant.”  

- It would be worthwhile to represent all the significant results in Tables or Figs in the main text and not in the Supplementary Materials. (line 204-209)    

Response: We have moved all the supplementary files with significant results to the main file, as figures or text.

- Unfortunately, I could not open the link of Supplementary Materials.     

Response: We are sorry for that, and hope that you will be able to open the link now. Most of the supplementary material has in the new version been moved to the main file.

- Does the duration of the AN or the time since recovery affect the results?    

Response: Thanks for this good suggestion. We added a correlation analysis between years since AN onset and SCFA plasma concentrations. We detect no significant correlations, see results line 237-239. We do unfortunately not have data on the exact time since recovery, we only know based on the inclusion criteria that it is at least 1 year.

- Were there any significant differences in the slopes of the correlations between BMI and SCFA concentrations between the AN subtypes? 

Response: Yes, these results were presented in the results text. P-values were found together with the slopes in a supplementary table. We now added the significant p-values to the results text (see line 232-236). All p-values are presented in Supplementary Table 2. R values are added in Figure 2.

- The Discussion is very detailed and thorough. Nice work.  

Response: We thank the reviewer for the encouraging words.

- The list of References is extensive, not all journal names are abbreviated.

Response: We used the Endnote reference style indicated in the instructions of Nutrients.

Reviewer 2 Report

The authors examined the associations between SCFAs plasma concentrations and anorexia nervosa (AN) and found that females with AN and AN_REC had lower plasma concentration of of butyric, isobutyric-, and isovaleric acid. The topic is very timely and interesting and would be of interest to broad readership of Nutrients. The manuscript is very well-written, however, there a a few point that I would like the authors to address.

1)     I cannot see any information regarding time of blood draw. This may introduce a potential confound if not controlled, hence, should be added as a limitation.

2)     I cannot see any information regarding dietary intake measurement. Energy intake, especially fibre intake and other covariates should be accounted for in the models. So, an explanation as to why the current study lacks these measures would be needed.

3)     I would prefer to see the full stats (t-test results, p values, Cohen’s d…etc.)r eported in the table 1 and 2 or in the text.

Author Response

The authors examined the associations between SCFAs plasma concentrations and anorexia nervosa (AN) and found that females with AN and AN_REC had lower plasma concentration of of butyric, isobutyric-, and isovaleric acid. The topic is very timely and interesting and would be of interest to broad readership of Nutrients. The manuscript is very well-written, however, there a a few point that I would like the authors to address.

Response: We thank the reviewer for the encouraging words, and for taking the time to review and improve our manuscript.

  • I cannot see any information regarding time of blood draw. This may introduce a potential confound if not controlled, hence, should be added as a limitation.

Response: Please see line 388-389  in limitations section “In addition, the sampling was not standardized for a specific time of the day, and participants were not instructed to be in a fasted or fed state.”

  • I cannot see any information regarding dietary intake measurement. Energy intake, especially fibre intake and other covariates should be accounted for in the models. So, an explanation as to why the current study lacks these measures would be needed.

Response: We added a sentence on this in the limitations section, line 401-402 “Unfortunately, we do not have access to dietary records.”

Please see also lines 398-400 “It is also possible that the dietary habits of the AN-REC group still differ from individuals without any history of an eating disorder [68] in ways that could influence the gut microbiota, e.g., the amount of ingested dietary fibers and resistant starch.”, and lines 412-415 “The optimal design would be a longitudinal study that followed individuals over the course of treatment through recovery that measured SCFA concentrations in plasma and feces combined with dietary records to decipher the effect of nutritional recovery post AN on the gut microbiota and its metabolites.”

  • I would prefer to see the full stats (t-test results, p values, Cohen’s d…etc.)r eported in the table 1 and 2 or in the text.

Response: We added the significant p-values related to the demographics data in table 1 to the results text (line 212-214). Since the results on differences between groups in plasma SCFA were presented both in table and figure, we decided to exclude former Table 2. and instead present these results only as box plots in Figure 1. All p-values are presented in the figure, the significant ones in bold.

Round 2

Reviewer 2 Report

Thanks for addressing my concerns.